# IMPLICIT AUTOENCODERS

## ABSTRACT

In this paper, we describe the "implicit autoencoder" (IAE), a generative autoencoder in which both the generative path and the recognition path are parametrized by implicit distributions. We use two generative adversarial networks to define the reconstruction and the regularization cost functions of the implicit autoencoder, and derive the learning rules based on maximum-likelihood learning. Using implicit distributions allows us to learn more expressive posterior and conditional likelihood distributions for the autoencoder. Learning an expressive conditional likelihood distribution enables the latent code to only capture the abstract and high-level information of the data, while the remaining information is captured by the implicit conditional likelihood distribution. For example, we show that implicit autoencoders can disentangle the global and local information, and perform deterministic or stochastic reconstructions of the images. We further show that implicit autoencoders can disentangle discrete underlying factors of variation from the continuous factors in an unsupervised fashion, and perform clustering and semi-supervised learning.

## 1 INTRODUCTION

Deep generative models have achieved remarkable success in recent years. One of the most successful models is the generative adversarial network (GAN) (Goodfellow et al., 2014), which employs a two player min-max game. The generative model, $G$, samples the noise vector $\mathbf{z} \sim p(\mathbf{z})$ and generates the sample $G(\mathbf{z})$. The discriminator, $D(\mathbf{x})$, is trained to identify whether a point $\mathbf{x}$ comes from the data distribution or the model distribution; and the generator is trained to maximally confuse the discriminator. The cost function of GAN is

$$\min_G \max_D \mathbb{E}_{\mathbf{x} \sim p_{\text{data}}}[\log D(\mathbf{x})] + \mathbb{E}_{\mathbf{z} \sim p(\mathbf{z})}[\log(1 - D(G(\mathbf{z})))]. \tag{1}$$

GANs can be viewed as a general framework for learning implicit distributions (Mohamed & Lakshminarayanan, 2016; Huszár, 2017). Implicit distributions are probability distributions that are obtained by passing a noise vector through a deterministic function that is parametrized by a neural network. In the probabilistic machine learning problems, implicit distributions trained with the GAN framework can learn distributions that are more expressive than the tractable distributions trained with the maximum-likelihood framework.

Variational autoencoders (VAE) (Kingma & Welling, 2014; Rezende et al., 2014) are another successful generative models that use neural networks to parametrize the posterior and the conditional likelihood distributions. Both networks are jointly trained to maximize a variational lower bound on the data log-likelihood. One of the limitations of VAEs is that they learn factorized distributions for both the posterior and the conditional likelihood distributions. In this paper, we propose the "implicit autoencoder" (IAE) that uses implicit distributions for learning more expressive posterior and conditional likelihood distributions. Learning a more expressive posterior will result in a tighter variational bound; and learning a more expressive conditional likelihood distribution will result in a global vs. local decomposition of information between the prior and the conditional likelihood. This enables the latent code to only capture the information that we care about such as the high-level and abstract information, while the remaining low-level information of data is separately captured by the noise vector of the implicit decoder.

Implicit distributions have been previously used in learning generative models in works such as adversarial autoencoders (AAE) (Makhzani et al., 2015), adversarial variational Bayes (AVB) (Mescheder

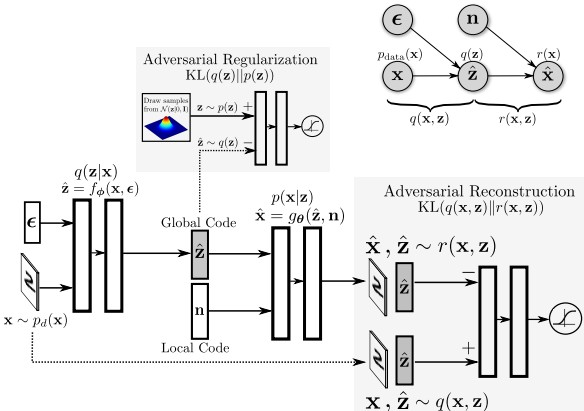

Figure 1: Architecture and graphical model of implicit autoencoders.

et al., 2017), ALI (Dumoulin et al., 2016), BiGAN (Donahue et al., 2016) and other works such as (Huszár, 2017; Tran et al., 2017). The global vs. local decomposition of information has also been studied in previous works such as PixelCNN autoencoders (van den Oord et al., 2016), PixelVAE (Gulrajani et al., 2016), variational lossy autoencoders (Chen et al., 2016b), PixelGAN autoencoders (Makhzani & Frey, 2017), or other works such as (Bowman et al., 2015; Graves et al., 2018; Alemi et al.). In the next section, we first propose the IAE and then establish its connections with the related works.

## 2 IMPLICIT AUTOENCODERS

Let $\mathbf{x}$ be a datapoint that comes from the data distribution $p_{\text{data}}(\mathbf{x})$. The encoder of the implicit autoencoder (Figure 1) defines an implicit variational posterior distribution $q(\mathbf{z}|\mathbf{x})$ with the function $\hat{\mathbf{z}} = f_{\phi}(\mathbf{x}, \boldsymbol{\epsilon})$ that takes the input $\mathbf{x}$ along with the *input noise* vector $\boldsymbol{\epsilon}$ and outputs $\hat{\mathbf{z}}$. The decoder of the implicit autoencoder defines an implicit conditional likelihood distribution $p(\mathbf{x}|\mathbf{z})$ with the function $\hat{\mathbf{x}} = g_{\theta}(\hat{\mathbf{z}}, \mathbf{n})$ that takes the code $\hat{\mathbf{z}}$ along with the *latent noise* vector $\mathbf{n}$ and outputs a reconstruction of the image $\hat{\mathbf{x}}$. In this paper, we refer to $\hat{\mathbf{z}}$ as the latent code or the *global code*, and refer to the latent noise vector $\mathbf{n}$ as the *local code*. Let $p(\mathbf{z})$ be a fixed prior distribution, $p(\mathbf{x}, \mathbf{z}) = p(\mathbf{z})p(\mathbf{x}|\mathbf{z})$ be the joint model distribution, and $p(\mathbf{x})$ be the model distribution. The variational distribution $q(\mathbf{z}|\mathbf{x})$ induces the *joint data distribution* $q(\mathbf{x}, \mathbf{z})$, the *aggregated posterior distribution* $q(\mathbf{z})$, and the *inverse posterior/encoder distribution* $q(\mathbf{x}|\mathbf{z})$ as follows:

$$q(\mathbf{x}, \mathbf{z}) = q(\mathbf{z}|\mathbf{x})p_{\text{data}}(\mathbf{x}) \qquad \hat{\mathbf{z}} \sim q(\mathbf{z}) = \int_{\mathbf{x}} q(\mathbf{x}, \mathbf{z})d\mathbf{x} \qquad q(\mathbf{x}|\mathbf{z}) = \frac{q(\mathbf{x}, \mathbf{z})}{q(\mathbf{z})} \qquad (2)$$

Maximum likelihood learning is equivalent to matching the model distribution $p(\mathbf{x})$ to the data distribution $p_{\text{data}}(\mathbf{x})$; and learning with variational inference is equivalent to matching the joint model distribution $p(\mathbf{x}, \mathbf{z})$ to the joint data distribution $q(\mathbf{x}, \mathbf{z})$. The entropy of the data distribution $\mathcal{H}_{\text{data}}(\mathbf{x})$, the entropy of the latent code $\mathcal{H}(\mathbf{z})$, the mutual information $\mathcal{I}(\mathbf{x}; \mathbf{z})$, and the conditional entropies $\mathcal{H}(\mathbf{x}|\mathbf{z})$ and $\mathcal{H}(\mathbf{z}|\mathbf{x})$ are all defined under the joint data distribution $q(\mathbf{x}, \mathbf{z})$ and its marginals $p_{\text{data}}(\mathbf{x})$ and $q(\mathbf{z})$. Using the aggregated posterior distribution $q(\mathbf{z})$, we can define the *joint reconstruction distribution* $r(\mathbf{x}, \mathbf{z})$ and the *aggregated reconstruction distribution* $r(\mathbf{x})$ as follows:

$$r(\mathbf{x}, \mathbf{z}) = q(\mathbf{z})p(\mathbf{x}|\mathbf{z}) \qquad \hat{\mathbf{x}} \sim r(\mathbf{x}) = \int_{\mathbf{z}} r(\mathbf{x}, \mathbf{z})d\mathbf{z} \qquad (3)$$

Note that in general we have $r(\mathbf{x}, \mathbf{z}) \neq q(\mathbf{x}, \mathbf{z}) \neq p(\mathbf{x}, \mathbf{z})$, $q(\mathbf{z}) \neq p(\mathbf{z})$, and $r(\mathbf{x}) \neq p_{\text{data}}(\mathbf{x}) \neq p(\mathbf{x})$.

We now use different forms of the aggregated evidence lower bound (ELBO) to describe the IAE and establish its connections with VAEs and AAEs.

$$\mathbb{E}_{\mathbf{x}\sim p_{\text{data}}(\mathbf{x})}[\log p(\mathbf{x})] \geq - \underbrace{\mathbb{E}_{\mathbf{x}\sim p_{\text{data}}(\mathbf{x})}\Big[\mathbb{E}_{q(\mathbf{z}|\mathbf{x})}[-\log p(\mathbf{x}|\mathbf{z})]\Big]}_{\text{VAE Reconstruction}} - \underbrace{\mathbb{E}_{\mathbf{x}\sim p_{\text{data}}(\mathbf{x})}\Big[\text{KL}(q(\mathbf{z}|\mathbf{x})\|p(\mathbf{z}))\Big]}_{\text{VAE Regularization}} \quad (4)$$

$$= - \underbrace{\mathbb{E}_{\mathbf{x}\sim p_{\text{data}}(\mathbf{x})}\Big[\mathbb{E}_{q(\mathbf{z}|\mathbf{x})}[-\log p(\mathbf{x}|\mathbf{z})]\Big]}_{\text{AAE Reconstruction}} - \underbrace{\text{KL}(q(\mathbf{z})\|p(\mathbf{z}))}_{\text{AAE Regularization}} - \underbrace{\mathcal{I}(\mathbf{z};\mathbf{x})}_{\text{Mutual Info.}} \quad (5)$$

$$= - \underbrace{\mathbb{E}_{\mathbf{z}\sim q(\mathbf{z})}\Big[\text{KL}(q(\mathbf{x}|\mathbf{z})\|p(\mathbf{x}|\mathbf{z}))\Big]}_{\text{IAE Reconstruction}} - \underbrace{\text{KL}(q(\mathbf{z})\|p(\mathbf{z}))}_{\text{IAE Regularization}} - \underbrace{\mathcal{H}_{\text{data}}(\mathbf{x})}_{\text{Entropy of Data}} \quad (6)$$

$$= - \underbrace{\text{KL}(q(\mathbf{x},\mathbf{z})\|r(\mathbf{x},\mathbf{z}))}_{\text{IAE Reconstruction}} - \underbrace{\text{KL}(q(\mathbf{z})\|p(\mathbf{z}))}_{\text{IAE Regularization}} - \underbrace{\mathcal{H}_{\text{data}}(\mathbf{x})}_{\text{Entropy of Data}} \quad (7)$$

See Appendix A for the proof. The standard formulation of the VAE (Equation 4) only enables us to learn factorized Gaussian posterior and conditional likelihood distributions. The AAE (Makhzani et al., 2015) (Equation 5) and the AVB (Mescheder et al., 2017) enable us to learn implicit posterior distributions, but their conditional likelihood distribution is still a factorized Gaussian distribution. However, the IAE enables us to learn implicit distributions for both the posterior and the conditional likelihood distributions. Similar to VAEs and AAEs, the IAE (Equation 6) has a reconstruction cost function and a regularization cost function, but trains each of them with a GAN. The IAE reconstruction cost is $\mathbb{E}_{\mathbf{z}\sim q(\mathbf{z})}\Big[\text{KL}(q(\mathbf{x}|\mathbf{z})\|p(\mathbf{x}|\mathbf{z}))\Big]$. The standard VAE uses a factorized decoder, which has a very limited stochasticity. Thus, the standard VAE performs almost *deterministic reconstructions* by learning to invert the deterministic mapping of the encoder. The IAE, however, uses a powerful implicit decoder to perform *stochastic reconstructions*, by learning to match the expressive decoder distribution $p(\mathbf{x}|\mathbf{z})$ to the inverse encoder distribution $q(\mathbf{x}|\mathbf{z})$. We note that there are other variants of VAEs that can also learn expressive decoder distributions by using autoregressive neural networks. We will discuss these models later in this section. Equation 8 contrasts the reconstruction cost of standard autoencoders that is used in VAEs/AAEs, with the reconstruction cost of IAEs.

$$\underbrace{\mathbb{E}_{\mathbf{x}\sim p_{\text{data}}(\mathbf{x})}\Big[\mathbb{E}_{q(\mathbf{z}|\mathbf{x})}[-\log p(\mathbf{x}|\mathbf{z})]\Big]}_{\text{AE Reconstruction}} = \underbrace{\mathbb{E}_{\mathbf{z}\sim q(\mathbf{z})}\Big[\text{KL}(q(\mathbf{x}|\mathbf{z})\|p(\mathbf{x}|\mathbf{z}))\Big]}_{\text{IAE Reconstruction}} + \underbrace{\mathcal{H}(\mathbf{x}|\mathbf{z})}_{\text{Cond. Entropy}} \quad (8)$$

We can see from Equation 8 that similar to IAEs, the reconstruction cost of the autoencoder encourages matching the decoder distribution to the inverse encoder distribution. But in autoencoders, the cost function also encourages minimizing the conditional entropy $\mathcal{H}(\mathbf{x}|\mathbf{z})$, or maximizing the mutual information $\mathcal{I}(\mathbf{x},\mathbf{z})$. Maximizing the mutual information in autoencoders enforces the latent code to capture both the global and local information. In contrast, in IAEs, the reconstruction cost does not penalize the encoder for losing the local information, as long as the decoder can invert the encoder distribution. In order to minimize the reconstruction cost function of the IAE, we re-write it in the form of a distribution matching cost function between the joint data distribution and the joint reconstruction distribution $\text{KL}(q(\mathbf{x},\mathbf{z})\|r(\mathbf{x},\mathbf{z}))$ (Equation 7). This KL divergence is approximately minimized with the *reconstruction GAN*. The IAE has also a regularization cost function $\text{KL}(q(\mathbf{z})\|p(\mathbf{z}))$ that matches the aggregated posterior distribution with a fixed prior distribution. This is the same regularization cost function used in AAEs (Equation 5), and is approximately minimized with the *regularization GAN*. Note that the last term in Equation 7 is the entropy of the data distribution that is fixed.

**Training Process.** We now describe the training process. We pass a given point $\mathbf{x}\sim p_{\text{data}}(\mathbf{x})$ through the encoder and the decoder to obtain $\hat{\mathbf{z}}\sim q(\mathbf{z})$ and $\hat{\mathbf{x}}\sim r(\mathbf{x})$. We now train the discriminator of the reconstruction GAN to identify the positive example $(\mathbf{x},\hat{\mathbf{z}})$ from the negative example $(\hat{\mathbf{x}},\hat{\mathbf{z}})$. Suppose this discriminator function at its optimality is $D^*(\mathbf{x},\mathbf{z})$. We try to confuse this discriminator by backpropagating through the negative example $(\hat{\mathbf{x}},\hat{\mathbf{z}})$ [1] and updating the encoder and decoder weights.

---

[1] We could also back-propagate through both the positive example $(\mathbf{x},\hat{\mathbf{z}})$ and the negative example $(\hat{\mathbf{x}},\hat{\mathbf{z}})$ to optimize the reconstruction cost. We observed that in the deterministic decoder case, it does not make any difference in the performance; but in the stochastic decoder case, it results in learning less useful representations, since it enables the encoder to change its weights freely in a way that is not compatible with the decoder.

More specifically, the generative loss of the reconstruction GAN is $T^*(\mathbf{x}, \mathbf{z}) = -\log D^*(\mathbf{x}, \mathbf{z})$, which defines the reconstruction cost of the IAE. We use the re-parametrization trick to update the encoder and decoder weights by computing the unbiased Monte Carlo estimate of the gradient of the reconstruction cost $T^*(\mathbf{x}, \mathbf{z})$ with respect to $(\boldsymbol{\phi}, \boldsymbol{\theta})$ as follows:

$$\nabla_{(\boldsymbol{\phi}, \boldsymbol{\theta})} \mathbb{E}_{r(\mathbf{x}, \mathbf{z})}\big[T^*(\mathbf{x}, \mathbf{z})\big] = \nabla_{(\boldsymbol{\phi}, \boldsymbol{\theta})} \mathbb{E}_{q_{\boldsymbol{\phi}}(\mathbf{z}) p_{\boldsymbol{\theta}}(\mathbf{x}|\mathbf{z})}\big[T^*(\mathbf{x}, \mathbf{z})\big] \tag{9}$$

$$= \mathbb{E}_{\mathbf{x} \sim p_{\text{data}}(\mathbf{x})} \mathbb{E}_{\boldsymbol{\epsilon}} \mathbb{E}_{\mathbf{n}}\big[\nabla_{(\boldsymbol{\phi}, \boldsymbol{\theta})} T^*(g_{\boldsymbol{\theta}}(f_{\boldsymbol{\phi}}(\mathbf{x}, \boldsymbol{\epsilon}), \mathbf{n}), f_{\boldsymbol{\phi}}(\mathbf{x}, \boldsymbol{\epsilon}))\big] \tag{10}$$

We call this process the *adversarial reconstruction*. Similarly, we train the discriminator of the regularization GAN to identify the positive example $\mathbf{z} \sim p(\mathbf{z})$ from the negative example $\hat{\mathbf{z}} \sim q(\mathbf{z})$. This discriminator now defines the regularization cost function, which can provide us with a gradient to update only the encoder weights. We call this process the *adversarial regularization*. Optimizing the adversarial regularization and reconstruction cost functions encourages $p(\mathbf{x}|\mathbf{z}) = q(\mathbf{x}|\mathbf{z})$ and $p(\mathbf{z}) = q(\mathbf{z})$, which results in the model distribution capturing the data distribution $p(\mathbf{x}) = p_{\text{data}}(\mathbf{x})$.

We note that in this work, we use the original formulation of GANs (Goodfellow et al., 2014) to match the distributions. As a result, the gradient that we obtain from the adversarial training, only approximately follows the gradient of the variational bound on the data log-likelihood. However, as shown in (Nowozin et al., 2016), the objective of the GAN can be modified to optimize any $f$-divergence including the KL divergence.

**Bits-Back Interpretation of the IAE Objective.** In Appendix B, we describe an information theoretic interpretation of the ELBO of IAEs (Equation 7) using the Bits-Back coding argument (Hinton & Van Camp, 1993; Chen et al., 2016b; Graves et al., 2018).

**Global vs. Local Decomposition of Information in IAEs.** In IAEs, the dimension of the latent vector along with its prior distribution defines the capacity of the latent code, and the dimension of the latent noise vector along with its distribution defines the capacity of the implicit decoder. By adjusting these dimensions and distributions, we can have a full control over the decomposition of information between the latent code and the implicit decoder. In one extreme case, by removing the noise vector, we can have a fully deterministic autoencoder that captures all the information by its latent code. In the other extreme case, we can remove the global latent code and have an unconditional implicit distribution that can capture the whole data distribution by itself. The global vs. local decomposition of information in IAEs is further discussed in Appendix C from an information theoretic perspective.

In IAEs, we can choose to only optimize the reconstruction cost or both the reconstruction and the regularization costs. In the following, we discuss four special cases of the IAE and establish connections with the related methods.

### 1. Deterministic Decoder without Regularization Cost

In this case, we remove the noise vectors from the IAE, which makes both $q(\mathbf{z}|\mathbf{x})$ and $p(\mathbf{x}|\mathbf{z})$ deterministic. We then only optimize the reconstruction cost $\mathbb{E}_{\mathbf{z} \sim q(\mathbf{z})}\Big[\text{KL}(q(\mathbf{x}|\mathbf{z})\|p(\mathbf{x}|\mathbf{z}))\Big]$. As a result, similar to the standard autoencoder, the deterministic decoder $p(\mathbf{x}|\mathbf{z})$ learns to match to the inverse deterministic encoder $q(\mathbf{x}|\mathbf{z})$, and thus the IAE learns to perform exact and deterministic reconstruction of the original image, while the latent code is learned in an unconstrained fashion. In other words, in standard autoencoders, the Euclidean cost *explicitly* encourages $\hat{\mathbf{x}}$ to reconstruct $\mathbf{x}$, and in case of uncertainty, performs mode averaging by blurring the reconstructions; however, in IAEs, the adversarial reconstruction *implicitly* encourages $\hat{\mathbf{x}}$ to reconstruct $\mathbf{x}$, and in case of uncertainty, captures this uncertainty by the local noise vector (Case 3), which results in sharp reconstructions.

### 2. Deterministic Decoder with Regularization Cost

In the previous case, the latent code was learned in an unconstrained fashion. We now keep the decoder deterministic and add the regularization term which matches the aggregated posterior distribution to a fixed prior distribution. In this case, the IAE reduces to the AAE with the difference that the IAE performs adversarial reconstruction rather than Euclidean reconstruction. This case of the IAE defines a valid generative model where the latent code captures all the information of the data distribution. In order to sample from this model, we first sample from the imposed prior $p(\mathbf{z})$ and then pass this sample through the deterministic decoder.

### 3. Stochastic Decoder without Regularization Cost

In this case of the IAE, we only optimize $\text{KL}(q(\mathbf{x}, \mathbf{z}) \| r(\mathbf{x}, \mathbf{z}))$, while $p(\mathbf{x}|\mathbf{z})$ is a stochastic implicit distribution. Matching the joint distribution $q(\mathbf{x}, \mathbf{z})$ to $r(\mathbf{x}, \mathbf{z})$ ensures that their marginal distributions would also match; that is, the aggregated reconstruction distribution $r(\mathbf{x})$ matches the data distribution $p_{\text{data}}(\mathbf{x})$. This model by itself defines a valid generative model in which both the prior, which in this case is $q(\mathbf{z})$, and the conditional likelihood $p(\mathbf{x}|\mathbf{z})$ are learned at the same time. In order to sample from this generative model, we initially sample from $q(\mathbf{z})$ by first sampling a point $\mathbf{x} \sim p_{\text{data}}(\mathbf{x})$ and then passing it through the encoder to obtain the latent code $\hat{\mathbf{z}} \sim q(\mathbf{z})$. Then we sample from the implicit decoder distribution conditioned on $\hat{\mathbf{z}}$ to obtain the stochastic reconstruction $\hat{\mathbf{x}} \sim r(\mathbf{x})$. If the decoder is deterministic (Case 1), the reconstruction $\hat{\mathbf{x}}$ would be the same as the original image $\mathbf{x}$. But if the decoder is stochastic, the global latent code only captures the abstract and high-level information of the image, and the stochastic reconstruction $\hat{\mathbf{x}}$ only shares this high-level information with the original $\mathbf{x}$. This case of the IAE is related to the PixelCNN autoencoder (van den Oord et al., 2016), where the decoder is parametrized by an autoregressive neural network which can learn expressive distributions, while the latent code is learned in an unconstrained fashion.

### 4. Stochastic Decoder with Regularization Cost

In the previous case, we showed that even without the regularization term, $r(\mathbf{x})$ will capture the data distribution. But the main drawback of the previous case is that its prior $q(\mathbf{z})$ is not a parametric distribution that can be easily sampled from. One way to fix this problem is to fit a parametric prior $p(\mathbf{z})$ to $q(\mathbf{z})$ once the training is complete, and then use $p(\mathbf{z})$ to sample from the model. However, a better solution would be to consider a fixed and pre-defined prior $p(\mathbf{z})$, and impose it on $q(\mathbf{z})$ during the training process. Indeed, this is the regularization term that the ELBO suggests in Equation 7. By adding the adversarial regularization cost function to match $q(\mathbf{z})$ to $p(\mathbf{z})$, we ensure that $r(\mathbf{x}) = p_{\text{data}}(\mathbf{x}) = p(\mathbf{x})$. Now sampling from this model only requires first sampling from the pre-defined prior $\mathbf{z} \sim p(\mathbf{z})$, and then sampling from the conditional implicit distribution to obtain $\hat{\mathbf{x}} \sim r(\mathbf{x})$. In this case, the information of data distribution is captured by both the fixed prior and the learned conditional likelihood distribution. Similar to the previous case, the latent code captures the high-level and abstract information, while the remaining local and low-level information is captured by the implicit decoder. We will empirically show this decomposition of information on different datasets in Section 2.1.1 and Section 2.1.2. This decomposition of information has also been studied in other works such as PixelVAE (Gulrajani et al., 2016), variational lossy autoencoders (Chen et al., 2016b), PixelGAN autoencoders (Makhzani & Frey, 2017) and variational Seq2Seq autoencoders (Bowman et al., 2015). However, the main drawback of these methods is that they all use autoregressive decoders which are not parallelizable, and are much more computationally expensive to scale up than the implicit decoders. Another advantage of implicit decoders to autoregressive decoders is that in implicit decoders, the local statistics is captured by the local code representation; but in autoregressive decoders, we do not learn a vector representation for the local statistics.

**Connections with ALI and BiGAN.** In ALI (Dumoulin et al., 2016) and BiGAN (Donahue et al., 2016) models, there are two separate networks that define the joint data distribution $q(\mathbf{x}, \mathbf{z})$ and the joint model distribution $p(\mathbf{x}, \mathbf{z})$. The parameters of these networks are trained using the gradient that comes from a single GAN that tries to match these two distributions. However, in the IAE, similar to VAEs or AAEs, the encoder and decoder are stacked on top of each other and trained jointly. So the gradient that the encoder receives comes through the decoder and the conditioning vector. In other words, in the ALI model, the input to the conditional likelihood is the samples of the prior distribution, whereas in the IAE, the input to the conditional likelihood is the samples of the variational posterior distribution, while the prior distribution is separately imposed on the aggregated posterior distribution by the regularization GAN. This makes the training dynamic of IAEs similar to that of autoencoders, which encourages better reconstructions. Recently, many variants of ALI have been proposed for improving its reconstruction performance. For example, the HALI (Belghazi et al., 2018) uses a Markovian generator to achieve better reconstructions, and ALICE (Li et al., 2017) augments the ALI's cost by a joint distribution matching cost function between $(\mathbf{x}, \hat{\mathbf{x}})$ and $(\mathbf{x}, \mathbf{x})$, which is different from our reconstruction cost.

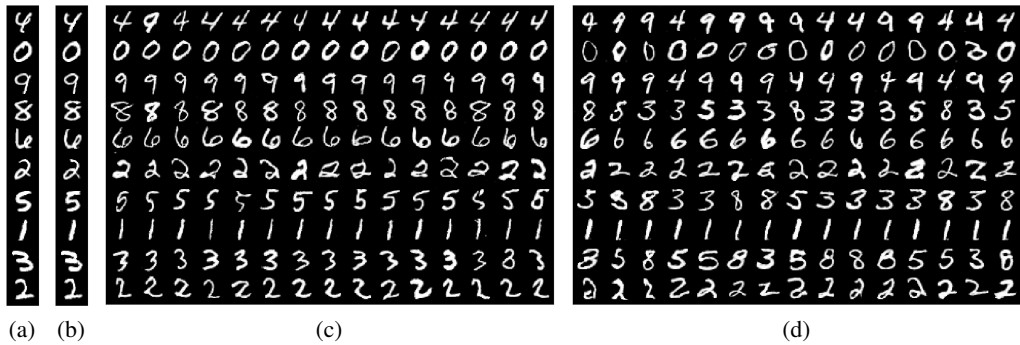

(a) (b)          (c)                       (d)

Figure 2: MNIST dataset. (a) Original images. (b) Deterministic reconstructions with 20D global vector. (c) Stochastic reconstructions with 10D global and 100D local vector. (d) Stochastic reconstructions with 5D global and 100D local vector.

## 2.1 EXPERIMENTS OF IMPLICIT AUTOENCODERS

### 2.1.1 GLOBAL VS. LOCAL DECOMPOSITION OF INFORMATION

In this section, we show that the IAE can learn a global vs. local decomposition of information between the latent code and the implicit decoder. We use the Gaussian distribution for both the global and local codes, and show that by adjusting the dimensions of the global and local codes, we can have a full control over the decomposition of information.

Figure 2 shows the performance of the IAE on the MNIST dataset. By removing the local code and using only a global code of size 20D (Figure 2b), the IAE becomes a deterministic autoencoder. In this case, the global code of the IAE captures all the information of the data distribution and the IAE achieves almost perfect reconstructions. By decreasing the global code size to 10D and using a 100D local code (Figure 2c), the global code retains the global information of the digits such as the label information, while the local code captures small variations in the style of the digits. By using a smaller global code of size 5D (Figure 2d), the encoder loses more local information and thus the global code captures more abstract information. For example, we can see from Figure 2d that the encoder maps visually similar digits such as $\{3, 5, 8\}$ or $\{4, 9\}$ to the same global code, while the implicit decoder learns to invert this mapping and generate stochastic reconstructions that share the

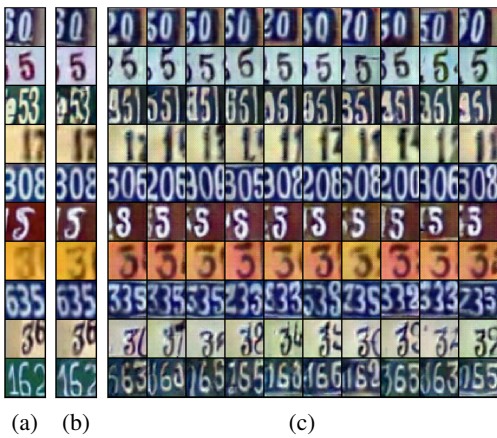
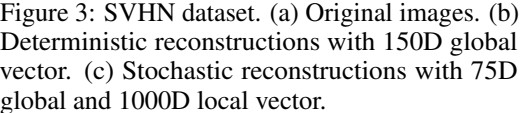

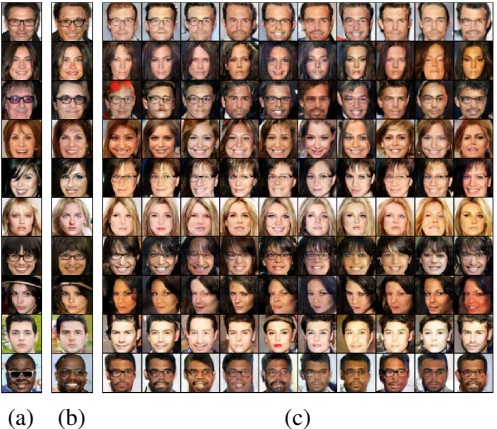

(a) (b)        (c)               (a) (b)        (c)

Figure 3: SVHN dataset. (a) Original images. (b) Deterministic reconstructions with 150D global vector. (c) Stochastic reconstructions with 75D global and 1000D local vector.

Figure 4: CelebA dataset. (a) Original images. (b) Deterministic reconstructions with 150D global vector. (c) Stochastic reconstructions with 50D global and 1000D local vector.

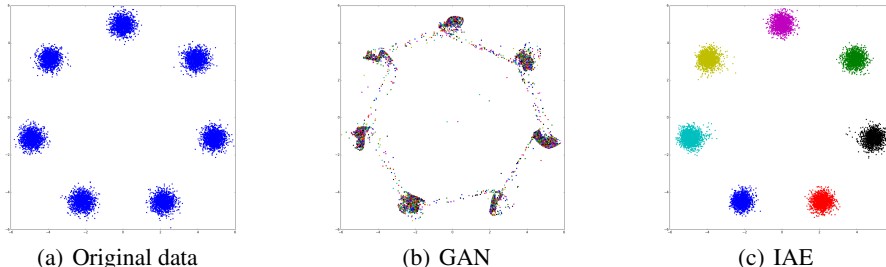

(a) Original data  (b) GAN  (c) IAE

Figure 5: Learning the mixture of Gaussian distribution by the standard GAN and the IAE.

same high-level information with the original images. Note that if we completely remove the global code, the local code captures all the information, similar to the standard unconditional GAN.

Figure 3 shows the performance of the IAE on the SVHN dataset. When using a 150D global code with no local code (Figure 3b), similar to the standard autoencoder, the IAE captures all the information by its global code and can achieve almost perfect reconstructions. However, when using a 75D global code along with a 1000D local code (Figure 3c), the global code of the IAE only captures the middle digit information as the global information, and loses the left and right digit information. At the same time, the implicit decoder learns to invert the encoder distribution by keeping the middle digit and generating synthetic left and right SVHN digits with the same style of the middle digit.

Figure 4 shows the performance of the IAE on the CelebA dataset. When using a 150D global code with no local code (Figure 4b), the IAE achieves almost perfect reconstructions. But when using a 50D global code along with a 1000D local code (Figure 4c), the global code of the IAE only retains the global information of the face such as the general shape of the face, while the local code captures the local attributes of the face such as eyeglasses, mustache or smile.

### 2.1.2 CLUSTERING AND SEMI-SUPERVISED LEARNING

In IAEs, by using a categorical global code along with a Gaussian local code, we can disentangle the discrete and continuous factors of variation, and perform clustering and semi-supervised learning.

**Clustering.** In order to perform clustering with IAEs, we change the architecture of Figure 1 by using a softmax function in the last layer of the encoder, as a continuous relaxation of the categorical global code. The dimension of the categorical code is the number of categories that we wish the data to be clustered into. The regularization GAN is trained directly on the continuous output probabilities of the softmax simplex, and imposes the categorical distribution on the aggregated posterior distribution. This adversarial regularization imposes two constraints on the encoder output. The first constraint is that the encoder has to make confident decisions about the cluster assignments. The second constraint is that the encoder must distribute the points evenly across the clusters. As a result, the global code only captures the discrete underlying factors of variation such as class labels, while the rest of the structure of the image is separately captured by the Gaussian local code of the implicit decoder.

Figure 5 shows the samples of the standard GAN and the IAE trained on the mixture of Gaussian data. Figure 5b shows the samples of the GAN, which takes a 7D categorical and a 10D Gaussian noise vectors as the input. Each sample is colored based on the one-hot noise vector that it was generated from. We can see that the GAN has failed to associate the categorical noise vector to different mixture components, and generate the whole data solely by using its Gaussian noise vector. Ignoring the categorical noise forces the GAN to do a continuous interpolation between different mixture components, which results in reducing the quality of samples. Figure 5c shows the samples of the IAE whose implicit decoder architecture is the same as the GAN. The IAE has a 7D categorical global code (inferred by the encoder) and a 10D Gaussian noise vector. In this case, the inference network of the IAE learns to cluster the data in an unsupervised fashion, while its generative path learns to condition on the inferred cluster labels and generate each mixture component using the stochasticity of the Gaussian noise vector. This example highlights the importance of using discrete latent variables for improving generative models. A related work is the InfoGAN (Chen et al., 2016a), which uses a reconstruction cost in the code space to prevent the GAN from ignoring the categorical noise vector. The relationship of InfoGANs with IAEs is discussed in details in Section 3.

Figure 6: Disentangling the content and style of the MNIST digits in an unsupervised fashion with implicit autoencoders. Each column shows samples of the model from one of the learned clusters. The style (local noise vector) is drawn from a Gaussian distribution and held fixed across each row.

Figure 6 shows the clustering performance of the IAE on the MNIST dataset. The IAE has a 30D categorical global latent code and a 10D Gaussian local code. Each column corresponds to the conditional samples from one of the learned clusters (only 20 are shown). The local code is sampled from the Gaussian distribution and held fixed across each row. We can see that the discrete global latent code of the network has learned discrete factors of variation such as the digit identities, while the writing style information is separately captured by the continuous Gaussian noise vector. This network obtains about 5% error rate in classifying digits in an unsupervised fashion, just by matching each cluster to a digit type.

**Semi-Supervised Learning.** The IAE can be used for semi-supervised classification. In order to incorporate the label information, we set the number of clusters to be the same as the number of class labels and additionally train the encoder weights on the labeled mini-batches to minimize the cross-entropy cost. On the MNIST dataset with 100 labels, the IAE achieves the error rate of 1.40%. In comparison, the AAE achieves 1.90%, and the Improved-GAN (Salimans et al., 2016) achieves 0.93%. On the SVHN dataset with 1000 labels, the IAE achieves the error rate of 9.80%. In comparison, the AAE achieves 17.70%, and the Improved-GAN achieves 8.11%.

## 3 FLIPPED IMPLICIT AUTOENCODERS

In this section, we describe the "Flipped Implicit Autoencoder" (FIAE), which is a generative model that is very closely related to IAEs. Let $\mathbf{z}$ be the latent code that comes from the prior distribution $p(\mathbf{z})$. The encoder of the FIAE (Figure 7) parametrizes an implicit distribution that uses the noise vector $\mathbf{n}$ to define the conditional likelihood distribution $p(\mathbf{x}|\mathbf{z})$. The decoder of the FIAE parametrizes an implicit distribution that uses the noise vector $\epsilon$ to define the variational posterior distribution $q(\mathbf{z}|\mathbf{x})$. In addition to the distributions defined in Section 2, we also define the *joint latent reconstruction distribution* $s(\mathbf{x}, \mathbf{z})$, and the *aggregated latent reconstruction distribution* $s(\mathbf{z})$ as follows:

$$s(\mathbf{x}, \mathbf{z}) = p(\mathbf{x})q(\mathbf{z}|\mathbf{x}) \qquad \hat{\mathbf{z}} \sim s(\mathbf{z}) = \int_{\mathbf{x}} s(\mathbf{x}, \mathbf{z})d\mathbf{x} \tag{11}$$

The objective of the standard variational inference is minimizing $\mathrm{KL}(q(\mathbf{x}, \mathbf{z})\|p(\mathbf{x}, \mathbf{z}))$, which is the variational upper-bound on $\mathrm{KL}(p_{\mathrm{data}}(\mathbf{x})\|p(\mathbf{x}))$. The objective of FIAEs is the reverse KL divergence $\mathrm{KL}(p(\mathbf{x}, \mathbf{z})\|q(\mathbf{x}, \mathbf{z}))$, which is the variational upper-bound on $\mathrm{KL}(p(\mathbf{x})\|p_{\mathrm{data}}(\mathbf{x}))$. The FIAE optimizes this variational bound by splitting it into a reconstruction term and a regularization

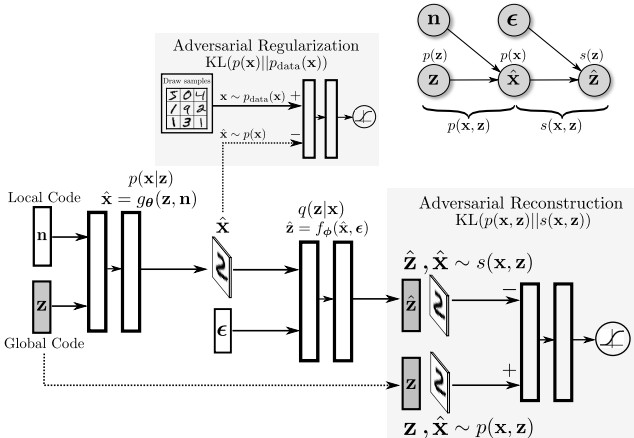

Figure 7: Architecture and graphical model of flipped implicit autoencoders.

term as follow:

$$\mathrm{KL}(p(\mathbf{x}) \| p_{\text{data}}(\mathbf{x})) \leq \underbrace{\mathrm{KL}(p(\mathbf{x}, \mathbf{z}) \| q(\mathbf{x}, \mathbf{z}))}_{\text{Variational Bound}} \tag{12}$$

$$= \underbrace{\mathrm{KL}(p(\mathbf{x}) \| p_{\text{data}}(\mathbf{x}))}_{\text{InfoGAN Regularization}} + \underbrace{\mathbb{E}_{\mathbf{z} \sim p(\mathbf{z})} \Big[ \mathbb{E}_{p(\mathbf{x}|\mathbf{z})} [-\log q(\mathbf{z}|\mathbf{x})] \Big]}_{\text{InfoGAN Reconstruction}} - \underbrace{\mathcal{H}(\mathbf{z}|\mathbf{x})}_{\text{Cond. Entropy}} \tag{13}$$

$$= \underbrace{\mathrm{KL}(p(\mathbf{x}) \| p_{\text{data}}(\mathbf{x}))}_{\text{FIAE Regularization}} + \underbrace{\mathbb{E}_{\mathbf{x} \sim p(\mathbf{x})} \Big[ \mathrm{KL}(p(\mathbf{z}|\mathbf{x}) \| q(\mathbf{z}|\mathbf{x})) \Big]}_{\text{FIAE Reconstruction}} \tag{14}$$

$$= \underbrace{\mathrm{KL}(p(\mathbf{x}) \| p_{\text{data}}(\mathbf{x}))}_{\text{FIAE Regularization}} + \underbrace{\mathrm{KL}(p(\mathbf{x}, \mathbf{z}) \| s(\mathbf{x}, \mathbf{z}))}_{\text{FIAE Reconstruction}} \tag{15}$$

where the conditional entropy $\mathcal{H}(\mathbf{z}|\mathbf{x})$ is defined under the joint model distribution $p(\mathbf{x}, \mathbf{z})$. Similar to the IAE, the FIAE has a regularization term and a reconstruction term (Equation 14 and Equation 15). The regularization cost uses a GAN to train the encoder (conditional likelihood) such that the model distribution $p(\mathbf{x})$ matches the data distribution $p_{\text{data}}(\mathbf{x})$. The reconstruction cost uses a GAN to train both the encoder (conditional likelihood) and the decoder (variational posterior) such that the joint model distribution $p(\mathbf{x}, \mathbf{z})$ matches the joint latent reconstruction distribution $s(\mathbf{x}, \mathbf{z})$.

**Connections with ALI and BiGAN.** In ALI (Dumoulin et al., 2016) and BiGAN (Donahue et al., 2016) models, the input to the recognition network is the samples of the real data $p_{\text{data}}(\mathbf{x})$; however, in FIAEs, the recognition network only gets to see the synthetic samples that come from the simulated data $p(\mathbf{x})$, while at the same time, the regularization cost ensures that the simulated data distribution is close the real data distribution. Training the recognition network on the simulated data in FIAEs is in spirit similar to the "sleep" phase of the wake-sleep algorithm (Hinton et al., 1995), during which the recognition network is trained on the samples that the network "dreams" up. One of the flaws of training the recognition network on the simulated data is that early in the training, the simulated data do not look like the real data, and thus the recognition path learns to invert the generative path in part of the data space that is far from the real data distribution. As the result, the reconstruction GAN might not be able to keep up with the moving simulated data distribution and get stuck in a local optimum. However, in our experiments with FIAEs, we did not find this to be a major problem.

**Connections with InfoGAN.** InfoGANs (Chen et al., 2016a), similar to FIAEs, train the variational posterior network on the simulated data; however, as shown in Equation 13, InfoGANs use an explicit reconstruction cost function (e.g., Euclidean cost) on the code space for learning the variational posterior. In order to compare FIAEs and InfoGANs, we train them on a toy dataset with four

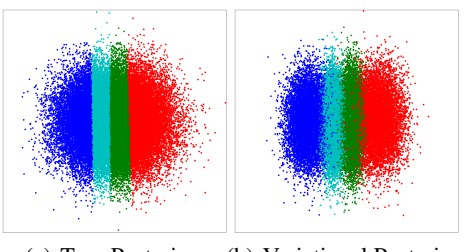

(a) True Posterior     (b) Variational Posterior

Figure 8: InfoGAN on a toy dataset. (a) True posterior. (b) Factorized Gaussian variational posterior.

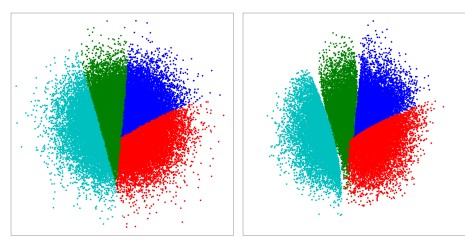

(a) True Posterior     (b) Variational Posterior

Figure 9: Flipped implicit autoencoder on a toy dataset. (a) True posterior. (b) Implicit variational posterior.

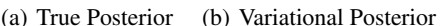

Figure 10: Reconstructions of the flipped implicit autoencoder on the MNIST dataset. Top row shows the MNIST test images, and bottom row shows the deterministic reconstructions.

data-points and use a 2D Gaussian prior (Figure 8 and Figure 9). Each colored cluster corresponds to the posterior distribution of one data-point. In InfoGANs, using the Euclidean cost to reconstruct the code corresponds to learning a factorized Gaussian variational posterior distribution (Figure 8b)[2]. This constraint on the variational posterior restricts the family of the conditional likelihoods that the model can learn by enforcing the generative path to learn a conditional likelihood whose true posterior could fit to the factorized Gaussian approximation of the posterior. For example, we can see in Figure 8a that the model has learned a conditional likelihood whose true posterior is axis-aligned, so that it could better match the factorized Gaussian variational posterior (Figure 8b). In contrast, the FIAE can learn an arbitrarily expressive variational posterior distribution (Figure 9b), which enables the generative path to learn a more expressive conditional likelihood and true posterior (Figure 9a).

One of the main flaws of optimizing the reverse KL divergence is that the variational posterior will have the mode-covering behavior rather than the mode-picking behavior. For example, we can see from Figure 8b that the Gaussian posteriors of different data-points in InfoGAN have some overlap; but this is less of a problem in the FIAE (Figure 9b), as it can learn a more expressive $q(\mathbf{z}|\mathbf{x})$. This mode-averaging behavior of the posterior can be also observed in the wake-sleep algorithm, in which during the sleep phase, the recognition network is trained using the reverse KL divergence objective.

The FIAE objective is not only an upper-bound on $\mathrm{KL}(p(\mathbf{x})\|p_{\mathrm{data}}(\mathbf{x}))$, but is also an upper-bound on $\mathrm{KL}(p(\mathbf{z})\|q(\mathbf{z}))$ and $\mathrm{KL}(p(\mathbf{z}|\mathbf{x})\|q(\mathbf{z}|\mathbf{x}))$. As a result, the FIAE matches the variational posterior $q(\mathbf{z}|\mathbf{x})$ to the true posterior $p(\mathbf{z}|\mathbf{x})$, and also matches the aggregated posterior $q(\mathbf{z})$ to the prior $p(\mathbf{z})$. For example, we can see in Figure 9b that $q(\mathbf{z})$ is very close to the Gaussian prior. However, the InfoGAN objective is theoretically not an upper-bound on $\mathrm{KL}(p(\mathbf{x})\|p_{\mathrm{data}}(\mathbf{x}))$, $\mathrm{KL}(p(\mathbf{z})\|q(\mathbf{z}))$ or $\mathrm{KL}(p(\mathbf{z}|\mathbf{x})\|q(\mathbf{z}|\mathbf{x}))$. As a result, in InfoGANs, the variational posterior $q(\mathbf{z}|\mathbf{x})$ need not be close to the true posterior $p(\mathbf{z}|\mathbf{x})$, or the aggregated posterior $q(\mathbf{z})$ does not have to match the prior $p(\mathbf{z})$.

### 3.1 Experiments of Flipped Implicit Autoencoders

**Reconstruction.** In this section, we show that the variational posterior distribution of the FIAE can invert its conditional likelihood function by showing that the network can perform reconstructions of the images. We make both the conditional likelihood and the variational posterior deterministic by removing both noise vectors $\mathbf{n}$ and $\epsilon$. Figure 10 shows the performance of the FIAE with a code size of 15 on the test images of the MNIST dataset. The reconstructions are obtained by first passing the image through the recognition network to infer its latent code, and then using the inferred latent code at the input of the conditional likelihood to generate the reconstructed image.

---

[2]In Figure 8b, we have trained both the mean and the standard deviation of the Gaussian posteriors.

**Clustering.** Similar to IAEs, we can use FIAEs for clustering. We perform an experiment on the MNIST dataset by choosing a discrete categorical latent code $\mathbf{z}$ of size 10, which captures the digit identity; and a continuous Gaussian noise vector $\mathbf{n}$ of size 10, which captures the style of the digit. The variational posterior distribution $q(\mathbf{z}|\mathbf{x})$ is also parametrized by an implicit distribution with a Gaussian noise vector $\epsilon$ of size 20, and performs inference only over the digit identity $\mathbf{z}$. Once the network is trained, we can use the variational posterior to cluster the test images of the MNIST dataset. This network achieves the error rate of about 2% in classifying digits in an unsupervised fashion by matching each categorical code to a digit type. We observed that when there is uncertainty in the digit identity, different draws of the noise vector $\epsilon$ results in different one-hot vectors at the output of the recognition network, showing that the implicit decoder can efficiently capture the uncertainty.

## 4 CONCLUSION

In this paper, we proposed the implicit autoencoder, which is a generative autoencoder that uses implicit distributions to learn expressive variational posterior and conditional likelihood distributions. We showed that in IAEs, the information of the data distribution is decomposed between the prior and the conditional likelihood. When using a low dimensional Gaussian distribution for the global code, we showed that the IAE can disentangle high-level and abstract information from the low-level and local statistics. We also showed that by using a categorical latent code, we can learn discrete factors of variation and perform clustering and semi-supervised learning.

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

## APPENDIX A    DERIVATION OF THE ELBO OF IMPLICIT AUTOENCODERS

$$\mathbb{E}_{\mathbf{x} \sim p_d(\mathbf{x})}[\log p(\mathbf{x})] \geq \mathbb{E}_{\mathbf{x} \sim p_d(\mathbf{x})}\left[\mathbb{E}_{q(\mathbf{z}|\mathbf{x})} \log \frac{p(\mathbf{x}, \mathbf{z})}{q(\mathbf{z}|\mathbf{x})}\right] \tag{16}$$

$$= \int q(\mathbf{x}, \mathbf{z}) \log \frac{p(\mathbf{x}, \mathbf{z})}{q(\mathbf{z}|\mathbf{x})} d\mathbf{x} d\mathbf{z} \tag{17}$$

$$= -\int q(\mathbf{x}, \mathbf{z}) \log \frac{q(\mathbf{z}|\mathbf{x})}{p(\mathbf{x}|\mathbf{z})} d\mathbf{x} d\mathbf{z} + \int q(\mathbf{z}) \log p(\mathbf{z}) d\mathbf{z} \tag{18}$$

$$= -\int q(\mathbf{x}, \mathbf{z}) \log \frac{q(\mathbf{z}|\mathbf{x})}{p(\mathbf{x}|\mathbf{z})} d\mathbf{x} d\mathbf{z} - \int p_d(\mathbf{x}) \log p_d(\mathbf{x}) d\mathbf{x} + \int q(\mathbf{z}) \log p(\mathbf{z}) d\mathbf{z} + \int p_d(\mathbf{x}) \log p_d(\mathbf{x}) d\mathbf{x} \tag{19}$$

$$= -\int q(\mathbf{x}, \mathbf{z}) \log \frac{p_d(\mathbf{x}) q(\mathbf{z}|\mathbf{x})}{p(\mathbf{x}|\mathbf{z})} d\mathbf{x} d\mathbf{z} + \int q(\mathbf{z}) \log p(\mathbf{z}) d\mathbf{z} - \mathcal{H}_{\text{data}}(\mathbf{x}) \tag{20}$$

$$= -\int q(\mathbf{x}, \mathbf{z}) \log \frac{q(\mathbf{x}, \mathbf{z})}{p(\mathbf{x}|\mathbf{z})} d\mathbf{x} d\mathbf{z} + \int q(\mathbf{z}) \log q(\mathbf{z}) d\mathbf{z} - \int q(\mathbf{z}) \log \frac{q(\mathbf{z})}{p(\mathbf{z})} d\mathbf{z} - \mathcal{H}_{\text{data}}(\mathbf{x}) \tag{21}$$

$$= -\int q(\mathbf{x}, \mathbf{z}) \log \frac{q(\mathbf{x}, \mathbf{z})}{q(\mathbf{z}) p(\mathbf{x}|\mathbf{z})} d\mathbf{x} d\mathbf{z} - \text{KL}(q(\mathbf{z}) \| p(\mathbf{z})) - \mathcal{H}_{\text{data}}(\mathbf{x}) \tag{22}$$

$$= -\text{KL}(q(\mathbf{x}, \mathbf{z}) \| r(\mathbf{x}, \mathbf{z})) - \text{KL}(q(\mathbf{z}) \| p(\mathbf{z})) - \mathcal{H}_{\text{data}}(\mathbf{x}) \tag{23}$$

## APPENDIX B    BITS-BACK INTERPRETATION OF THE IAE OBJECTIVE

In this section, we describe an information theoretic interpretation of the ELBO of IAEs (Equation 7) using the Bits-Back coding argument (Hinton & Van Camp, 1993; Chen et al., 2016b; Graves et al., 2018). Maximizing the variational lower bound is equivalent to minimizing the expected description length of a source code for the data distribution $p_{\text{data}}(\mathbf{x})$ when the code is designed under the model distribution $p(\mathbf{x})$. In order to transmit $\mathbf{x}$, the sender uses a two-part code. It first transmits $\mathbf{z}$, which ideally would only require $\mathcal{H}(\mathbf{z})$ bits; however, since the code is designed under $p(\mathbf{z})$, the sender has to pay the penalty of $\text{KL}(q(\mathbf{z}) \| p(\mathbf{z}))$ extra bits to compensate for the mismatch between $q(\mathbf{z})$ and $p(\mathbf{z})$. After decoding $\mathbf{z}$, the receiver now has to resolve the uncertainty of $q(\mathbf{x}|\mathbf{z})$ in order to reconstruct $\mathbf{x}$, which ideally requires the sender to transmit the second code of the length $\mathcal{H}(\mathbf{x}|\mathbf{z})$ bits. However, since the code is designed under $p(\mathbf{x}|\mathbf{z})$, the sender has to pay the penalty of $\mathbb{E}_{\mathbf{z} \sim q(\mathbf{z})}\left[\text{KL}(q(\mathbf{x}|\mathbf{z}) \| p(\mathbf{x}|\mathbf{z}))\right]$ extra bits on average to compensate for the fact that the conditional decoder $p(\mathbf{x}|\mathbf{z})$ has not perfectly captured the inverse encoder distribution $q(\mathbf{x}|\mathbf{z})$; i.e., the autoencoder has failed to achieve perfect stochastic reconstruction. But for a given $\mathbf{x}$, the sender could use the stochasticity of $q(\mathbf{z}|\mathbf{x})$ to encode other information. Averaged over the data distribution, this would get the sender $\mathcal{H}(\mathbf{z}|\mathbf{x})$ "bits back" that needs to be subtracted in order to find the true cost for transmitting $\mathbf{x}$:

$$\ell_{\mathcal{C}_{\text{IAE}}} = \mathcal{H}(\mathbf{z}) + \text{KL}(q(\mathbf{z}) \| p(\mathbf{z})) + \mathcal{H}(\mathbf{x}|\mathbf{z}) + \text{KL}(q(\mathbf{x}, \mathbf{z}) \| r(\mathbf{x}, \mathbf{z})) - \mathcal{H}(\mathbf{z}|\mathbf{x}) \tag{24}$$

$$= \mathcal{H}_{\text{data}}(\mathbf{x}) + \text{KL}(q(\mathbf{x}, \mathbf{z}) \| r(\mathbf{x}, \mathbf{z})) + \text{KL}(q(\mathbf{z}) \| p(\mathbf{z})) \tag{25}$$

From Equation 25, we can see that the IAE only minimizes the extra number of bits required for transmitting $\mathbf{x}$, while the VAE minimizes the total number of bits required for the transmission.

**Continuous Variables.** The Bits-Back argument is also applicable to continuous random variables. Suppose $\mathbf{x}$ and $\mathbf{z}$ are real-valued random variables. Let $h(\mathbf{x})$ and $h(\mathbf{z})$ be the differential entropies of $\mathbf{x}$ and $\mathbf{z}$; and $\mathcal{H}(\mathbf{x})$ and $\mathcal{H}(\mathbf{z})$ be the discrete entropies of the quantized versions of $\mathbf{x}$ and $\mathbf{z}$, with the quantization interval of $\Delta \mathbf{x}$ and $\Delta \mathbf{z}$. We have

$$\mathcal{H}(\mathbf{x}) \to h(\mathbf{x}) - \log \Delta \mathbf{x}, \quad \text{as } \Delta \mathbf{x} \to 0 \tag{26}$$

$$\mathcal{H}(\mathbf{z}) \to h(\mathbf{z}) - \log \Delta \mathbf{z}, \quad \text{as } \Delta \mathbf{z} \to 0 \tag{27}$$

The sender first transmits the real-valued random variable $\mathbf{z}$, which requires transmission of $\mathcal{H}(\mathbf{z}) = h(\mathbf{z}) - \log \Delta \mathbf{z}$ bits, as well as $\text{KL}(q(\mathbf{z}) \| p(\mathbf{z}))$ extra bits. As $\Delta \mathbf{z} \to 0$, we will have $\mathcal{H}(\mathbf{z}) \to \infty$, which, as expected, implies that the sender would need infinite number of bits to source code and send the real-valued random variable $\mathbf{z}$. However, as we shall see, we are going to get most of these bits back from the receiver at the end. After the first message, the sender then sends the second message, which requires transmission of $h(\mathbf{x}|\mathbf{z}) - \log \Delta \mathbf{x}$ bits, as well as $\mathbb{E}_{\mathbf{z} \sim q(\mathbf{z})}\left[\text{KL}(q(\mathbf{x}|\mathbf{z}) \| p(\mathbf{x}|\mathbf{z}))\right]$ extra

bits. Once the receiver decodes $\mathbf{z}$, and form that decodes $\mathbf{x}$, it can decode a secondary message of the average length $\mathcal{H}(\mathbf{z}|\mathbf{x}) = h(\mathbf{z}|\mathbf{x}) - \log \Delta \mathbf{z}$, which needs to be subtracted in order to find the true cost for transmitting $\mathbf{x}$:

$$
\begin{aligned}
\ell_{\mathcal{C}_{\text{IAE}}} &= h(\mathbf{z}) - \log \Delta \mathbf{z} + \text{KL}(q(\mathbf{z})\|p(\mathbf{z})) + h(\mathbf{x}|\mathbf{z}) - \log \Delta \mathbf{x} + \text{KL}(q(\mathbf{x}, \mathbf{z})\|r(\mathbf{x}, \mathbf{z})) - (h(\mathbf{z}|\mathbf{x}) - \log \Delta \mathbf{z}) \\
&= h(\mathbf{x}) - \log \Delta \mathbf{x} + \text{KL}(q(\mathbf{x}, \mathbf{z})\|r(\mathbf{x}, \mathbf{z})) + \text{KL}(q(\mathbf{z})\|p(\mathbf{z})) \\
&= \mathcal{H}_{\text{data}}(\mathbf{x}) + \text{KL}(q(\mathbf{x}, \mathbf{z})\|r(\mathbf{x}, \mathbf{z})) + \text{KL}(q(\mathbf{z})\|p(\mathbf{z}))
\end{aligned}
\tag{28}
$$

From Equation 28, we can interpret the IAE cost as the extra number of bits required for the transmission of $\mathbf{x}$.

## APPENDIX C  GLOBAL VS. LOCAL DECOMPOSITION OF INFORMATION IN IMPLICIT AUTOENCODERS

In IAEs, the global code (prior) captures the global information of data, while the remaining local information is captured by the local noise vector (conditional likelihood). In this section, we describe the global vs. local decomposition of information from an information theoretic perspective. In order to transmit $\mathbf{x}$, the sender first transmits $\mathbf{z}$ and then transmits the residual bits required for reconstructing $\mathbf{x}$, using a source code that is designed based on $p(\mathbf{x}, \mathbf{z})$. If $p(\mathbf{z})$ and $p(\mathbf{x}|\mathbf{z})$ are powerful enough, in theory, they can capture any $q(\mathbf{x}, \mathbf{z})$, and thus regardless of the decomposition of information, the sender would only need to send $\mathcal{H}_{\text{data}}(\mathbf{x})$ bits. In this case, the ELBO does not prefer one decomposition of information to another. But if the capacities of $p(\mathbf{z})$ and $p(\mathbf{x}|\mathbf{z})$ are limited, the sender will have to send extra bits due to the distribution mismatch, resulting in the regularization and reconstruction errors. But now different decompositions of information will result in different numbers of extra bits. So the sender has to decompose the information in a way that is compatible with the source codes that are designed based on $p(\mathbf{z})$ and $p(\mathbf{x}|\mathbf{z})$. The prior $p(\mathbf{z})$ that we use in this work is a low-dimensional Gaussian or categorical distribution. So the regularization cost encourages the sender to encode low-dimensional or simple concepts in $\mathbf{z}$ that is consistent with $p(\mathbf{z})$; otherwise, the sender would need to pay a large cost for $\text{KL}(q(\mathbf{z})\|p(\mathbf{z}))$. The choice of the information encoded in $\mathbf{z}$ would also affect the extra number of bits of $\mathbb{E}_{\mathbf{z} \sim q(\mathbf{z})}\Big[\text{KL}(q(\mathbf{x}|\mathbf{z})\|p(\mathbf{x}|\mathbf{z}))\Big]$, which is the reconstruction cost. This is because the conditional decoder $p(\mathbf{x}|\mathbf{z})$ with its limited capacity is supposed to capture the inverse encoder distribution $q(\mathbf{x}|\mathbf{z})$. So the sender must encode the kind of information in $\mathbf{z}$ that after being observed, can maximally remove the stochasticity of $q(\mathbf{x}|\mathbf{z})$ so as to lower the burden on $p(\mathbf{x}|\mathbf{z})$ for matching to $q(\mathbf{x}|\mathbf{z})$. So the reconstruction cost encourages learning the kind of concepts that can remove as much uncertainty as possible from the data distribution. By balancing the regularization and reconstruction costs, the latent code learns global concepts which are low-dimensional or simple concepts that can maximally remove uncertainty from data. Examples of global concepts are digit identities in the MNIST dataset, objects in natural images or topics in documents.

## APPENDIX D  IMPLEMENTATION DETAILS

### D.1  GLOBAL CODE CONDITIONING

There are two methods to implement how the reconstruction GAN conditions on the global code.

**Location-Dependent Conditioning.** Suppose the size of the first convolutional layer of the discriminator is `(batch, width, height, channels)`. We use a one layer neural network with 1000 ReLU hidden units to transform the global code of size `(batch, global_code_size)` to a spatial tensor of size `(batch, width, height, 1)`. We then broadcast this tensor across the channel dimension to get a tensor of size `(batch, width, height, channels)`, and then add it to the first layer of the discriminator as an adaptive bias. In this method, the latent vector has spatial and location-dependent information within the feature map. This is the method that we used in deterministic and stochastic reconstruction experiments.

**Location-Invariant Conditioning.** Suppose the size of the first convolutional layer of the discriminator is `(batch, width, height, channels)`. We use a linear mapping to transform the

global code of size (batch, global_code_size) to a tensor of size (batch, channels). We then broadcast this tensor across the width and height dimensions, and then add it to the first layer of the discriminator as an adaptive bias. In this method, the global code is encouraged to learn the global information that is location-invariant such as the class label information. We used this method in all the clustering and semi-supervised learning experiments.

### D.2 Network Architectures for Deterministic and Stochastic Reconstructions

The regularization discriminator in all the experiments is a two-layer neural network, where each layer has 2000 hidden units with the ReLU activation function. The architecture of the encoder, the decoder and the reconstruction discriminator for each dataset is as follows.

**MNIST:**

| Encoder | Decoder | Disc. Reconstruction GAN |
|---|---|---|
| $\mathbf{x} \in \mathbb{R}^{28 \times 28}$ | $\hat{\mathbf{z}} \in \mathbb{R}^{20}$ and $\mathbf{n} \in \mathbb{R}^{100}$ | $(\mathbf{x}, \hat{\mathbf{z}})$ or $(\hat{\mathbf{x}}, \hat{\mathbf{z}})$ |
| FC. 2000 ReLU. | FC. 1024 ReLU. BN | $4 \times 4$ Conv. 64 ReLU. Stride 2. BN |
| FC. 2000 ReLU. | FC. $128 \times 7 \times 7$ ReLU. BN | $4 \times 4$ Conv. 128 ReLU. Stride 2. BN |
| FC. 20 Linear. BN | $4 \times 4$ UpConv. 64 ReLU. Stride 2. BN | FC. 1024 ReLU. BN |
| | $4 \times 4$ UpConv. 1 Sigmoid. Stride 2. | FC. 1 Linear |

**SVHN:**

| Encoder | Decoder | Disc. Reconstruction GAN |
|---|---|---|
| $\mathbf{x} \in \mathbb{R}^{32 \times 32 \times 3}$ | $\hat{\mathbf{z}} \in \mathbb{R}^{75}$ and $\mathbf{n} \in \mathbb{R}^{1000}$ | $(\mathbf{x}, \hat{\mathbf{z}})$ or $(\hat{\mathbf{x}}, \hat{\mathbf{z}})$ |
| $4 \times 4$ Conv. 64 ReLU. Stride 2. BN | FC. $256 \times 4 \times 4$ ReLU. BN | $4 \times 4$ Conv. 64 ReLU. Stride 2. BN |
| $4 \times 4$ Conv. 128 ReLU. Stride 2. BN | $4 \times 4$ UpConv. 128 ReLU. Stride 2. BN | $4 \times 4$ Conv. 128 ReLU. Stride 2. BN |
| $4 \times 4$ Conv. 256 ReLU. Stride 2. BN | $4 \times 4$ UpConv. 64 ReLU. Stride 2. BN | $4 \times 4$ Conv. 256 ReLU. Stride 2. BN |
| FC. 75 Linear. BN | $4 \times 4$ UpConv. 3 Tanh. Stride 2. | FC. 1 Linear |

**CelebA:**

| Encoder | Decoder | Disc. Reconstruction GAN |
|---|---|---|
| $\mathbf{x} \in \mathbb{R}^{48 \times 48 \times 3}$ | $\hat{\mathbf{z}} \in \mathbb{R}^{50}$ and $\mathbf{n} \in \mathbb{R}^{1000}$ | $(\mathbf{x}, \hat{\mathbf{z}})$ or $(\hat{\mathbf{x}}, \hat{\mathbf{z}})$ |
| $6 \times 6$ Conv. 128 ReLU. Stride 2. BN | FC. $512 \times 6 \times 6$ ReLU. BN | $6 \times 6$ Conv. 128 ReLU. Stride 2. BN |
| $6 \times 6$ Conv. 256 ReLU. Stride 2. BN | $6 \times 6$ UpConv. 256 ReLU. Stride 2. BN | $6 \times 6$ Conv. 256 ReLU. Stride 2. BN |
| $6 \times 6$ Conv. 512 ReLU. Stride 2. BN | $6 \times 6$ UpConv. 128 ReLU. Stride 2. BN | $6 \times 6$ Conv. 512 ReLU. Stride 2. BN |
| FC. 50 Linear. BN | $6 \times 6$ UpConv. 3 Tanh. Stride 2. | FC. 1 Linear |

