# OpenReview forum: "Implicit Autoencoders"
_ICLR.cc/2019/Conference_

### Official Review · AnonReviewer2 · 2018-11-03
**Increasing the expressiveness of decoder by an implicit decoder looks interesting, and it enables the decompositions of high-level abstract information from low one.**

**Rating:** 6
**Confidence:** 3

**Review:**

The paper proposed an implicit auto-encoder, featuring both the encoder and decoder constituted by implicit distributions. Adversary training is used train the models, similar to the technique used in the AVB model. The main difference with AVB is the use of an implicit decoder, which endows the model with the ability to disentangle the data into high-level abstract representation and local representation. Although sharing some similarities, the extension of using implicit decoder is interesting, and leading to some interesting results.

My main concern on this paper is the lack of any quantitive results to compare with other similar models. We only see the model can do some task, but cannot assess how well it did comparing to other models.

---

> ### Author Response · Authors · 2018-11-15
> **Rebuttal**
>
> We thank the reviewer for the feedback.
> The main claim of our paper is that IAEs can learn useful unsupervised representations using their latent code. This is because the latent code of the IAE can only focus on capturing high-level abstractions, while the remaining low-level information is separately captured by the implicit decoder. While we have done many qualitative experiments to support our claim, we believe that the best currently available method to quantitatively evaluate unsupervised representations is to evaluate them on downstream tasks. So we quantitatively evaluated the usefulness of the IAE representations in our clustering and semi-supervised learning experiments on the MNIST and the SVHN datasets, and showed that the IAE can achieve very competitive results (Section 2.1.2 and Section 3.1). However, we believe it is important for the generative modeling research community to find better metrics for evaluating the quality of unsupervised representations in generative models.

---

> ### Author Response · Authors · 2018-12-12
> **Follow up**
>
> We thank the reviewer again for the feedback. We were wondering if our rebuttal addressed the concerns of the reviewer.

---

### Official Review · AnonReviewer3 · 2018-11-06
**Interesting models (with a potentially indifferent encoder?)**

**Rating:** 6
**Confidence:** 4

**Review:**

The paper presents two generative autoencoding models, that optimize a variational objective by adversarial training of implicit distributions. Applications in generative modeling, clustering, semi-supervised learning and disentangling “global” vs “local” variations in data are presented.

In particular, the first model, called implicit autoencoder, maximizes the ELBO using two adversarial objectives: an adversarial regularizer penalizes the deviation of the aggregated posterior from the prior and the adversarial reconstruction penalizes the disagreement between the joint distribution and the joint reconstruction. Since both implicit distributions use a noise source, they can both explain variations in the data. The paper argues (and presents experimental results to suggest) that the global vs local information is the separation in these two components. The second architecture, called _flipped_ implicit autoencoder replaces the role of code and input in the first architecture, changing the training objective to reverse KL-divergence. The relationship between the proposed models to several prior works including ALI, BiGAN, AVB, adversarial autoencoders, wake-sleep, infoGAN is discussed.

The paper is nicely written, and the theory part seems to be sound. The strength of this paper is that it ties together a variety of different autoencoding generative architectures. In particular, I found it interesting that AVB and InfoGAN become special cases of the regular and flipped model, where an implicit term in the loss is replaced by an explicit likelihood term.

I have some issues/questions (did I miss something obvious?):

1) A mode of failure: suppose the encoder simply produces a garbled code that does not reflect that data manifold, yet it has the right aggregated posterior. In this case, the implicit generator and discriminator should ignore the code. However, the generator can still match the joint reconstruction and to the correct joint distribution. The loss can go towards its minimum. Note that this is not a problem in AVB.

2) Global vs local info: a closely related issue to previous failure mode is that encoder has no incentive to produce informative codes. While the paper argues that the size of the code can decide the decomposition of local vs. global information, even for a wide bottleneck in the autoencoder, the code can have no (or very little) information.

3) Could you please elaborate on the footnotes of page 3?

---

> ### Author Response · Authors · 2018-11-15
> **Rebuttal**
>
> We thank the reviewer for the positive feedback.
>
> 1,2) We have a detailed discussion about the global vs. local decomposition of information in Appendix C. In the case of having a wide bottleneck and a powerful implicit decoder p(x|z), the ELBO does not prefer one decomposition of information to another. So in this case, as the reviewer points out, in theory, the network can capture all the information solely by the latent code, solely by the implicit decoder, or by a combination of them. However, empirically, it is the *dynamic of training/optimization* and not the objective that determines the decomposition of information. The architecture of the IAE is very similar to that of the standard autoencoder. In fact, if we remove the local latent code, the IAE becomes a deterministic autoencoder, and the network learns the same kind of high-level concepts that an autoencoder would learn. However, in the presence of the local code, as shown by our empirical experiments, the network tries to capture as much information as possible by the latent code, but instead of averaging over the remaining information and generating blurry images, it captures the distribution of the remaining information by the implicit decoder.
>
> 3) In order to match r(x,z) to q(x,z), we back-propagate the reconstruction discriminator gradient through negative examples. In this case, we are considering q(x,z) as the target distribution that r(x,z) is trained to match to. The process of learning r(x,z) requires learning the encoder, which indirectly changes the underlying target distribution q(x,z) to a new target distribution q(x,z). In other words, r(x,z) is aiming a moving target distribution, which is q(x,z), and once the reconstruction discriminator is confused, we will have r(x,z)=q(x,z). We empirically observed that by only back-propagating through negative examples, we will provide a more stable target distribution for r(x,z) to aim for, which results in a more stable training dynamic and better empirical performance.

---

> ### Author Response · Authors · 2018-12-12
> **Response to the updated review**
>
> We noticed that the reviewer has reduced the rating without modifying the review. We were wondering if there is any new concern that we can address.

---

### Official Review · AnonReviewer1 · 2018-11-06
**Experiments in paper do not implement the objective in paper.**

**Rating:** 3
**Confidence:** 3

**Review:**

This paper introduces the implicit autoencoder, which purports to be a VAE with an implicit encoding and decoding distribution.

My principle problem with the paper and reason for my strong rejection is that there appears to be a complete separation between the discussion and theory of the paper and the actual experiments run.  The paper's discussion and theory all centers around rewriting the ordinary ELBO lower bound on the marginal likelihood in equations (4) through (7) where it is shown that this can be recast in the form of two KL divergences, one between the representational joint q(x,z) = p_data(x) encoder(z|x) and the 'reconstruction joint' r(x,z) = encoder_marginal(z) decoder(x|z), and one between the encoding marginal q(z) and the generative prior p(z).   The entire text of the paper then discusses the similarities between this formulation of the objective and some of the alternatives as well as discussing how this objective might behave in various limits.

However, this is not the objective that is actually trained. In the  "Training Process" section is it revealed that an ordinary GAN discriminator is trained.  The ordinary GAN objective does not minimize a KL divergence, it is a minimax formulation of a Jensen Shannon divergence as the original GAN paper notes.  More specifically, you can optimize a KL divergence with a GAN, as shown in the f-GAN paper (1606.00709) but this requires attention be paid to the functional form of the loss and structure of the discriminator.  No such care was taken in this case.  As such the training process does not minimize the objective derived or discussed.  Not to mention that in practice a further hack is employed wherein only the negative example passes gradients to the generator.

While is not specified in the training process section, assuming the ordinary GAN objective (Equation 1) is used, according to their own reference (AVB) the optimal decoder should be:  D = 1/(1 + r(z,x)/q(z,x))  for which we have that what they deem the 'generative loss of the reconstruction GAN' is T = log(1 + r(z,x)/q(z,x))  .   When we take unbiased gradients of the expectation of this quantity, we do not obtain an unbiased gradient of the KL divergence between q(z,x) and r(z,x).

Throughout the paper, factorized Gaussian distributions are equated with tractable variational approximations.  While it is common to use a mean field gaussian distribution for the decoder in VAEs this is by no means required.  Many papers have investigated the use of more powerful autoregressive or flow based decoders, as this paper itself cites (van der Oord et al. 2016).  The text further misrepresents the current literature when it claims that the IAE uniquely "generalizes the idea of deterministic reconstruction to stochastic reconstruction by learning a decoder distribution that learns to match to the inverse encoder distribution".  All VAEs have employ stochastic reconstruction, if the authors again here meant to distinguish a powerful implicit decoder from a mean field gaussian one, the choice of language here is wrong.

Given that there are three joint distributions in equation (the generative model, the representational joint and the reconstruction joint), the use of Conditional entropy H(x|z) and mutual information I(x, z) are ambiguous.  While the particular joint distribution is implied by context in the equations, please spell it out for the reader.

The "Global vs. Local Decomposition of Information in IAEs" section conflates dimensionality with information capacity.  While these are likely correlated for real neural networks, at least fundamentally an arbitrary amount of information could be stored in even a 1 dimensional continuous random variable.  This is not addressed.

The actual experiments look nice, its just that objective used to train the resulting networks is not the one presented in the paper.

------

In light of the author's response I am changing my review from 2 to 3.  I still feel as though the paper should be rejected.  While I appreciate that there is a clear history of using GANs to target otherwise intractable objectives, I still feel like those papers are all very explicit about the fact that they are modifying the objective when they do so.  I find this paper confusing and at times erroneous.  The added appendix on the bits back argument for instance I believe is flawed.

"It first transmits z, which ideally would only require H(z) bits; however, since the code is designed
under p(z), the sender has to pay the penalty of KL(q(z)kp(z)) extra bits"

False.  The sender is not trying to send an unconditional latent code, they are trying to send the code for a given image, z \sim q(z|x).  Under usual communication schemes this would be sent via an entropic code designed for the shared prior at the cost of the cross entropy \int q(z|x) \log p(z) and the excess bits would be KL(q(z|x) | p(z)), not Kl(q(z)|p(z)).

The appendix ends with "IAE only minimizes the extra number of bits required for transmitting x, while the VAE minimizes the total number of bits required for the transmission" but the IAE = VAE by Equation (4-6).  They are equivalent, how can one minimizing something the other doesn't?  In general the paper to me reads at times as VAE=IAE but IAE is better.  While it very well might be true that the objective trained in the paper (a joint GAN objective attempting to minimize the Jensen Shannon divergence between both  (1) the joint data density q(z,x) and the aggregated reconstruction density r(z,x) and (2) the aggregated posterior q(z) and the prior p(z)) is better than a VAE (as the experiments themselves suggest), the rhetoric of the paper suggests that the IAE referred to throughout is Equation (6).  Equation 6 is equivalent to a VAE.

I think the paper would greatly benefit from a rewriting of the central story.  The paper has a good idea in it, I just feel as though it is not well presented in its current form and worry that if accepted in this form might cause more confusion than clarity.  Combined with what I view as some technical flaws especially in the appendices I still must vote for a rejection.

---

> ### Author Response · Authors · 2018-11-13
> **Rebuttal (Part 1)**
>
> We thank the reviewer for the feedback.
>
> The reviewer has a principle concern about one of the arguments of our paper, which has resulted in a very strong negative feedback about the whole paper. This argument is that the KL divergence between two distributions p(x) and q(x) can be *approximately* minimized by training a GAN that tries to match these two distributions. More specifically, the theoretical contribution of our paper is to re-derive the ELBO as the summation of two KL divergences and a fixed term: -KL(q(z)||p(z))-KL(q(x,z)||r(x,z))-H_data. These KL divergences are not tractable to optimize, so the empirical contribution of our paper is to show that we can use GANs to approximately minimize each of these KL divergences, and that the resulting *empirical* algorithm can perform useful tasks such as variational inference, clustering or semi-supervised learning.
>
> Firstly, we would like to point out that, theoretically, the standard GAN optimizes the JS divergence, which is a symmetric divergence, whose square root is a metric. In the minimization of the JS divergence, we try to get the two distributions as close as possible, which almost always results in the minimization of the KL divergence. There could be some pathological cases where the KL divergence does not decrease, but we empirically show that this approximate optimization works in our experiments. That being said, we never claimed that we are exactly optimizing the ELBO, and in several parts of the paper we have explicitly pointed out that we are only approximately optimizing the ELBO. For example, in the paper, we mention that "This KL divergence is *approximately* minimized with the reconstruction GAN", or that "This is the same regularization cost function used in AAEs, and is *approximately* minimized with the regularization GAN".
>
> Secondly, replacing an intractable divergence with another tractable divergence is a common idea used in many generative models such as adversarial autoencoders, the wake-sleep algorithm or ALI/BiGANs. In fact, the adversarial autoencoder (Eq. 5) uses the *exact same approximation* that we use, by replacing the intractable KL(q(z)||p(z)) in the code space with the adversarial cost. The concern of the reviewer can be similarly raised for the adversarial autoencoder, as it does not exactly optimizes the ELBO; nevertheless, the adversarial autoencoder shows that this approximation results in a useful generative model. Another example is the wake-sleep algorithm, in which the wake-phase optimizes the right KL divergence of data and model, but the sleep phase replaces this KL divergence with the reversed KL divergence and optimizes that instead. As the result, the wake-sleep algorithm does not exactly optimizes the ELBO; nevertheless, it is very successful in training sigmoid belief networks. Another example is the ALI/BiGAN methods which use the JS divergence in their formulations, but recently many papers such as [1] have argued that these methods are *approximately* optimizing the ELBO.
> Similar to all these works, in implicit autoencoders, by replacing intractable KL divergences with the adversarial training, we are not exactly optimizing the ELBO; nevertheless, we empirically show that the adversarial training can actually match the distributions that the KL divergence aims to match; and this results in a useful and practical algorithm. For example, the IAE can successfully learn expressive variational posteriors that can almost perfectly match the true posteriors (Fig. 9) using adversarial training; the IAE can empirically achieve very competitive clustering and semi-supervised learning results (Section 2.1.2); and can perform useful tasks such as high-level vs. low-level decomposition of information (Fig. 2,3,4).
>
> Thirdly, as the reviewer points out, the objective of the GAN can be modified to optimize any f-divergence including the KL divergence. Thus, in theory, by replacing the KL divergences of the IAE with the f-GAN objectives, we can very closely follow the gradient of the ELBO. Indeed, in the initial phases of this project, we did perform some experiments with the f-GAN objective, but observed that its empirical performance is very similar to the original GAN objective. This result has also been independently reported in many other works including the original f-GAN paper, which reports that "all three divergences [JS, KL and Hellinger] produce equally realistic samples". So given that the f-divergence objective did not bring any empirical benefit in our case, we chose to perform the experiments of this project with the standard GAN objective.
>
> Finally, We revised our paper by adding a paragraph to fully discuss this issue and to better clarify both our theoretical and empirical contributions. We hope that this revision of the paper along with the above response address the main concern of the reviewer.
>
> [1] Ferenc Huszár. Variational inference using implicit distributions, 2017.

---

> ### Author Response · Authors · 2018-11-13
> **Rebuttal (Part 2)**
>
> Reviewer: "Not to mention that in practice a further hack is employed wherein only the negative example passes gradients to the generator."
>
> In order to match r(x,z) to q(x,z), we back-propagate the reconstruction discriminator gradient through negative examples. In this case, we are considering q(x,z) as the target distribution that r(x,z) is trained to match to. The process of learning r(x,z) requires learning the encoder, which indirectly changes the underlying target distribution q(x,z) to a new target distribution q(x,z). In other words, r(x,z) is aiming a moving target distribution, which is q(x,z), and once the reconstruction discriminator is confused, we will have r(x,z)=q(x,z). We empirically observed that by only back-propagating through negative examples, we will provide a more stable target distribution for r(x,z) to aim for, which results in a more stable training dynamic and better empirical performance.
> ==============
> Reviewer: "Many papers have investigated the use of more powerful autoregressive or flow-based decoders, as this paper itself cites (van der Oord et al. 2016)."
>
> We have extensively discussed these autoregressive-decoder models in several parts of the paper (page 4 and 5), and have cited all the related works such as "PixelVAE", "Variational Lossy Autoencoders", "PixelGAN", "Fixing the broken ELBO" and "Associative compression networks". In page 5, for example, we mention the advantages of implicit-decoder based models to autoregressive-decoder based models by explaining that implicit decoders can scale to larger images, are less computationally expensive, and can capture vector representations for the local statistics.
> ==============
> Reviewer: "The text further misrepresents the current literature when it claims that the IAE uniquely generalizes the idea of deterministic reconstruction to stochastic reconstruction by learning a decoder distribution that learns to match to the inverse encoder distribution."
>
> This statement was made in the context of comparing the reconstruction cost of IAEs with VAEs. The reconstruction cost of IAEs, unlike that of VAEs, explicitly tries to match the decoder distribution p(x|z) to the inverse encoder distribution q(x|z). We revised our paper and modified this statement based on the reviewer's suggestions.
> ==============
> Reviewer: "the use of Conditional entropy H(x|z) and mutual information I(x, z) are ambiguous."
>
> In the page 2 of the paper, these terms have been explicitly defined: "The entropy of the data distribution H(x), the entropy of the latent code H(z), the mutual information I(x;z), and the conditional entropies H(x|z) and H(z|x) are all defined under the joint data distribution q(x,z) and its marginals p_data(x) and q(z)."
> ==============
> Reviewer: "The "Global vs. Local Decomposition of Information in IAEs" section conflates dimensionality with information capacity. While these are likely correlated for real neural networks, at least fundamentally an arbitrary amount of information could be stored in even a 1 dimensional continuous random variable. This is not addressed."
>
> We updated the paper by adding a section explaining how the Bits-Back argument is also applicable to continuous random variables in neural networks.

---

> ### Author Response · Authors · 2018-12-12
> **Response to the updated review**
>
> We thank the reviewer for updating the review. We try to further clarify additional points that were brought up in the updated review.
> ==========
> "While I appreciate that there is a clear history of using GANs to target otherwise intractable objectives. I still feel like those papers are all very explicit about the fact that they are modifying the objective when they do so."
>
> We have explicitly mentioned the objective of IAE and how it is optimized with GANs in several parts of the paper. We appreciate any suggestions on how we can make this more explicit.
> ==========
> "I find this paper confusing and at times erroneous. The added appendix on the bits back argument for instance I believe is flawed."
>
> As we describe below, we believe the arguments of the paper that the reviewer is referring to are correct.
> ==========
> "False. The sender is not trying to send an unconditional latent code, they are trying to send the code for a given image."
>
> We believe both the reviewer's and our derivation of the bits-back argument are correct. The reviewer's derivation corresponds to the VAE form of the ELBO, and ours correspond to the IAE form of the ELBO. In the bits-back argument, we are trying to transmit the data-distribution to the receiver. The x and z come from the joint data-distribution q(x,z)=p_data(x)q(z|x)=q(z)q(x|z), and we use a source code designed under the joint model distribution p(x,z)=p(z)p(x|z). We can construct the two-part code in two different ways, depending on how we sample from q(x,z). In the reviewer's derivation (VAE form), we use the equation q(x,z)=p_data(x)q(z|x), and sample from q(x,z) by first sampling x from the data-distribution and then sampling z from the conditional q(z|x). We then encode *conditional* z using p(z) which requires E_q(z|x)[-log p(z)] bits, and after deducting the bits-backs H(z|x), we send E_q(x,z)[-log p(z)] - H(z|x) = E_x KL(q(z|x)||p(z)) bits, as the reviewer points out. After that, we send the remaining bits in the second message. However, there is another way to sample from q(x,z) and construct the two-part code, which corresponds to the IAE form of the ELBO. In this method, we use the equation q(x,z)=q(z)q(x|z), and sample from q(x,z) by first sampling *unconditionally* from the aggregated posterior q(z) and then sampling from the conditional q(x|z). In this case, the first message of the two-part code only encodes the *unconditional* sample z from the marginal q(z) using the source code designed under p(z) which requires KL(q(z)||p(z)) extra bits, and the second message corresponds to the bits required to encode the uncertainty of q(x|z). These two methods of sampling from q(x,z) and constructing the two-part codes are two different interpretations of the same equation, and both are correct. One results in the VAE form of the ELBO, and the other results in the IAE form of the ELBO.
> ==========
> "The appendix ends with "IAE only minimizes the extra number of bits required for transmitting x, while the VAE minimizes the total number of bits required for the transmission" but the IAE = VAE by Equation (4-6) ... They are equivalent, how can one minimizing something the other doesn't?"
>
> Actually, VAE = IAE + H_data. The VAE objective corresponds to the total number of bits, meaning that the optimal value for the VAE objective is the entropy of data. But the IAE objective corresponds to the extra number of bits, meaning that the optimal value for the IAE objective is zero, which corresponds to the case where the model distribution is equal to the data distribution. Since H_data is fixed, optimizing the IAE objective also optimizes the VAE objective, but the value of these objectives are not equal.
> ==========
> "In general the paper to me reads at times as VAE=IAE but IAE is better ... Equation 6 is equivalent to a VAE."
>
> There are many ways to re-write the ELBO. The VAE objective is one form of writing the ELBO, and the IAE objective is another form of writing the ELBO. The advantage of writing the ELBO in form of Equation 6 is that this equation tells us that the ELBO is equal to the summation of two distribution-matching objectives. Now by replacing these distribution-matching objectives with the GAN objectives, we can learn implicit distributions for both the posterior and the conditional likelihood distributions.
> The underlying objective of VAEs and IAEs is the same, but just because two methods optimize the same objective, does not mean they learn equally good generative models. For example, the normalizing flow VAE or AVBs optimize the same ELBO that the VAE optimizes, but they can learn more expressive posterior distributions, which results in better distribution-matching between the variational and true posteriors, tighter variational bound and thus better generative models. Similarly, the IAE can learn more expressive posterior and conditional likelihoods, which results in better distribution-matching in the IAE objective, and thus better generative models.

---

### Public Comment · ~Mingzhang_Yin1 · 2018-12-21
**Related references**

I appreciate the proposed IAE method which uses implicit distribution for both encoder and decoder!

I would also like to point out some related works using implicit distributions in the variational inference/VAE that may serve as proper comparisons.

https://arxiv.org/abs/1805.11183
https://arxiv.org/abs/1705.10119
https://arxiv.org/abs/1810.02789

---

### Meta-Review · Area_Chair1 · 2018-12-19
**Interesting idea whose presentation could be less confusing**

**Confidence:** 3
**Recommendation:** Reject

**Metareview:**

The paper proposes an original idea for training a generative model based on an objective inspired by a VAE-like evidence lower bound (ELBO), reformulated as two KL terms, which are then approximately optimized by two GANs. They thus use implicit distributions for both the posterior and the conditional likelihood. The idea is original and intriguing. But reviewers and AC found that the paper currently suffered from the following weaknesses: a) The presentation of the approach is unclear, due primarily to the fact that it doesn't throughout unambiguously enough separate the VAE-like ELBO *inspiration*, from what happens when replacing the two KL terms by GANs, i.e. the actual algorithm used. This is a big conceptual jump that would deserve being discussed and analyzed more carefully and thoroughly. b) Reviewers agreed that the paper does not sufficiently evaluate the approach in comparative experiments with alternatives, in particular its generative capabilities, in addition to the provided evaluations of the learned representation on downstream tasks.
Reviewers did not reach a clear consensus on this paper, although discussion led two of them to revise their assessment score slightly towards each other's. One reviewer judged the paper currently too confusing (point a) putting more weight on this aspect than the other reviewers.
Based on the paper and the review discussion thread, the AC judges that while it is an original, interesting and potentially promising approach, its presentation can and should be much clarified and improved.